# Evaluation of a modified quantitative polymerase chain reaction assay for genus *Schistosoma* detection using stool and urine samples from schistosomiasis endemic areas in Kenya

**Henry Kanyi**[1]*, **Richelle W. Kihoro**[1], **Benard Chieng**[1], **Sylvie Araka**[1], **Henry Emisiko**[1], **Thierry Ramos**[2], **Sarah Nogaro**[2], **Sammy M. Njenga**[1]

1 Eastern and Southern Africa Centre of International Parasite Control (ESACIPAC), Kenya Medical Research Institute (KEMRI), Nairobi, Kenya, 2 FIND, Geneva, Switzerland

* hkanyi@kemri.go.ke

**Data Availability Statement:** Kindly note that the consent forms utilized to collect data from participants explicitly stated that access to the data

## Abstract

### Introduction

The microscopy-based Kato-Katz and urine filtration techniques have traditionally faced challenges in the detection of schistosomiasis in areas with low infection levels. A modified singleplex *Schistosoma* genus-specific quantitative real-time polymerase chain reaction (qPCR) assay was therefore evaluated as a sensitive and confirmatory schistosomiasis diagnostic test.

### Methodology

The qPCR assay utilized primers and probe targeting internal transcribed spacer– 2 (ITS2) sequence of *S. mansoni*, *S. haematobium* and *S. intercalatum*. A plasmid (pDMD801, 100pg/ul) was used as an internal amplification control and its qPCR assays were run in parallel to the *Schistosoma* assays. This assay utilized samples collected from 774 participants and microscopically examined for three consecutive days. A total of 699 day-one samples (urine and stools) from two schistosomiasis endemic sites were analyzed. Similarly, 75 persons from a non-endemic control site provided both urine and stool samples that were also analyzed.

### Results

Using microscopy, the proportion of positives in the two endemic regions altogether was 289/699 (41.3%). Using qPCR, 50.4% of the samples (352/699) were found to be positive for schistosome infection. The percentage of positive samples was slightly higher at 57.8% (203/351) in the *S. mansoni* endemic site compared with the *S. haematobium* site at 42.8% (149/348). Majority of the microscopy results were light infections at 26.8% (n = 94) and 26.1% (n = 91) while qPCR majority of the infections were high at 41.6% (n = 146) and

is limited to study staff, representatives from FIND and KEMRI, members of the ethics committee and the regulatory authorities. Therefore, access to anonymized and de-identified data can be obtained by emailing the Kenya Medical Research Institute (KEMRI) Scientific Ethics Review Unit (SERU) at seru@kemri.go.ke.

**Funding:** This work was supported by the Bill and Melinda Gates Foundation (grant award: INV_024908) that was awarded to SH. The funder had no role in decision to publish, or preparation of the manuscript.

**Competing interests:** The authors declare that they have no competing interests.

**Abbreviations:** CAA, Cathodic anodic antigen; CCA, Circulating Cathodic Antigen; Ct, Cycle threshold; DNA, deoxyribonucleic acid; EPG, Eggs per gram; ITS-2, Internal transcribed spacer– 2; KEMRI, Kenya Medical Research Institute; M, Microscopy; NPV, Negative predictive values; qPCR, Quantitative Polymerase Chain Reaction; PBS, Phosphate Buffered Saline; PCI, Posterior credible interval; PPV, Positive predictive values; PVP, Polyvinylpyrrolidone; Se, Sensitivity; Sp, Specificity; SERU, Scientific and Ethics Review Unit; UCP-LF, : Up-Converting Phosphor Lateral Flow; WHO, World Health Organization.

31.3% (n = 109) for the *S. mansoni* and *S. haematobium* sites, respectively. There were no positives detected by either microscopy or qPCR in the non-endemic site. Using Bayesian Latent Class Model, which does not use any technique as a gold standard, qPCR showed higher sensitivity (86.4% (PCI: 82.1–90.3)) compared to microscopy (75.6% (PCI: 71.1–80.0)).

## Conclusions

This study documents a single day-one sample modified *Schistosoma* qPCR assay as a powerful improved molecular assay for the detection of schistosomiasis infection that utilize either stool or urine samples. The assay is therefore recommended for monitoring in areas with low infection levels to enable accurate determination of the disease's control endpoint.

## Introduction

Schistosomiasis (also known as bilharzia) is a prevalent parasitic disease that often leads to chronic illness (if left untreated) especially amongst populations living in areas that lack adequate safe water and sanitation facilities [1]. The human infection is caused by 6 trematode species, namely *Schistosoma guineensis*, *S. haematobium*, *S. intercalatum*, *S. mansoni*, *S. japonicum* and *S. mekongi*. In Sub-Saharan Africa, the predominant species are *S. haematobium* and *S. mansoni* that cause urogenital and intestinal schistosomiasis, respectively [2]. The infection is prevalent in 78 countries, with about 240 million people infected globally, thus making it a major public health concern. Sub-Saharan Africa bears the greatest burden of the disease accounting for 93% of the global infection, of which children and young adults are the most affected groups [3, 4]. Humans are infected when they come into contact with fresh water infested with the larval forms of the parasite known as cercariae, which penetrates the skin. In the body, the larvae develop into adult schistosomes. The adult worms live in the veins that drain from the urinary tract and intestines [4]. If left untreated, schistosomiasis could lead to anemia, stunted growth, reduced mental ability and chronic inflammation of the organs, which can be fatal in the most serious instances [2].

The control of schistosomiasis has primarily been done through mass drug administration (MDA) campaigns of praziquantel which is the primary drug recommended by the World Health Organization for chemopreventive therapy [5]. The drug has been shown to be easily administrable, highly effective, tolerable and has minimal side effects [6]. Efforts led by the Kenya's National School Based Deworming Programme (NSBDP), implemented in 2012, have shown some impact on the overall prevalence of schistosomiasis after repeated rounds of MDAs [7, 8].

Success of control and elimination of any infection is highly dependent on the availability of highly sensitive and specific diagnostic techniques. Diagnosis of schistosomiasis has traditionally relied on urine filtration microscopy and the Kato-Katz methods for the detection of urogenital and intestinal infections, respectively. These techniques are widely used especially in resource poor settings because they are relatively cheap, simple, specific and easily applicable in the field setting [9–12]. However, they have the disadvantage of being time-consuming, laborious, requiring well-trained and experienced personnel, and have poor sensitivity, especially in the context of low-intensity infection [4, 11]. While the search for accurate, widely adaptable and user-friendly new diagnostic tools for schistosomiasis is gaining momentum,

the World Health Organization (WHO) target product profile (TPP) guidelines recommend that the ideal requirements for clinical sensitivity and clinical specificity of the test tools should be above 75% and 96.5% respectively for a sample size of 100 individuals [4]. Whilst the microscopy-based urine filtration and Kato-Katz techniques fulfil some of the attributes of the TPPs, they do not consistently meet the sensitivity requirements outlined in the TPP. These guidelines continue to guide the search for new tools to support monitoring and evaluation of schistosomiasis control efforts.

Lately, the development of tests that detect circulating cathodic antigen (CCA) and circulating anodic antigen (CAA) that are secreted by living schistosomes seems to be adding impetus to the development of field-friendly and easily applicable diagnostic tools [13, 14]. A test for CAA is available as a lab-based lateral flow format–known as the Up-Converting Phosphor Lateral Flow (UCP-LF) CAA assay. The UCP-LF CAA assay has shown its usefulness and high sensitivity for the diagnosis of *S. haematobium* and *S. mansoni* in low endemicity settings. However, this test is not field friendly and can only be deployed for use in a central laboratory. Additionally, being a commercially available kit, its adoption for use in a resource scarce setting in future may be limited by the associated costs and lot-to-lot variations [15, 16]. Customization of available tests for use in diverse environmental settings would be a boost to schistosomiasis elimination efforts especially in the low resource endemic countries.

The application of various polymerase chain reaction (PCR) techniques such as conventional PCR, real-time quantitative PCR (qPCR) as well as droplet digital PCR (ddPCR) have facilitated the diagnosis of schistosomiasis [17–19]. Real-time quantitative polymerase chain reaction (qPCR) assay for the detection of the *Schistosoma* deoxyribonucleic acids (DNA) in stool, urine and serum samples is also considered a highly sensitive and specific tool for monitoring of schistosomiasis control/elimination programmes. Several studies have demonstrated the superior performance of qPCR on single species of *Schistosoma* or while utilizing a single type of sample [9, 20–25]. Elsewhere, the technique has been advanced leading to the development of qPCR assay that simultaneously detect *Schistosoma* DNA and an internal control [25]. Unlike conventional PCR assays, the use of real-time qPCR has proven advantageous because it can quantitatively detect fewer copies of target DNA, and does not require running of the gel electrophoresis to visualize DNA thus is less laborious [24].

Given that the qPCR can quantify DNA, it is therefore possible to use its cycle threshold (Ct) value as a proxy indicator for parasite burden at a glance [21, 24]. Despite this superior performance, the use of qPCR is generally limited due to the requirement of highly skilled manpower, sophisticated equipment and costly reagents. However, in the context of low parasite burden in areas undergoing elimination, it remains one of the most ideal diagnostic tool for evaluating treatment end point alongside the other gold standard techniques [22, 23, 26–28].. . .

Unlike the previous PCR assays done for genus *Schistosoma*, the current assay varies in the following aspects. First, it was used to test both stool and urine samples from the respective schistosomiasis (SCH) endemic and non-endemic sites on a single platform. Secondly, the assay utilized a plasmid as an internal control for DNA extraction but the sample and the internal control assays were done separately (singleplex). Finally, the performance of the assay was validated using same person's stool and urine samples collected from schistosomiasis non-endemic site. As a result, this modified singleplex *Schistosoma* genus-specific qPCR assay was therefore evaluated as a sensitive and confirmatory schistosomiasis diagnostic test for possible use in accurate determination of the disease's control endpoint.

## Materials and methods

### Ethics statement

The study received ethical clearance from the Scientific and Ethics Review Unit (SERU) of KEMRI (KEMRI/SERU protocol No. 4057). Permission for sample collection was further sought and obtained from the respective county governments. Community mobilization and sensitization was done through meetings organized by the local administrators at the village level. The study participants provided written informed consent for sample collection which included that further molecular analysis would be conducted on the samples. For participants less than 18 years old, parental consent was obtained followed by assent from children (13-<18 years) before enrollment. Illiterate participants provided a thumbprint on the consent form and the consent form was also signed by an impartial witness.

### Study design and study area

This was a comparative diagnostic study to evaluate a modified qPCR diagnostic test using stored samples. These samples were collected from two sites that had been identified based on schistosomiasis prevalence and species (Fig 1). The areas are also undergoing control interventions through MDA being steered by the Neglected Tropical Diseases (NTDs) programme in the Ministry of Health (MoH), Kenya. The first site was South Sakwa Ward in Bondo Sub-county, Siaya County which is endemic for *S. mansoni* [29, 30]. The village borders the shores of Lake Victoria in the western Kenya region. The main economic activity of this area is predominantly fishing and subsistence farming. The area is characterized by warm, dry and

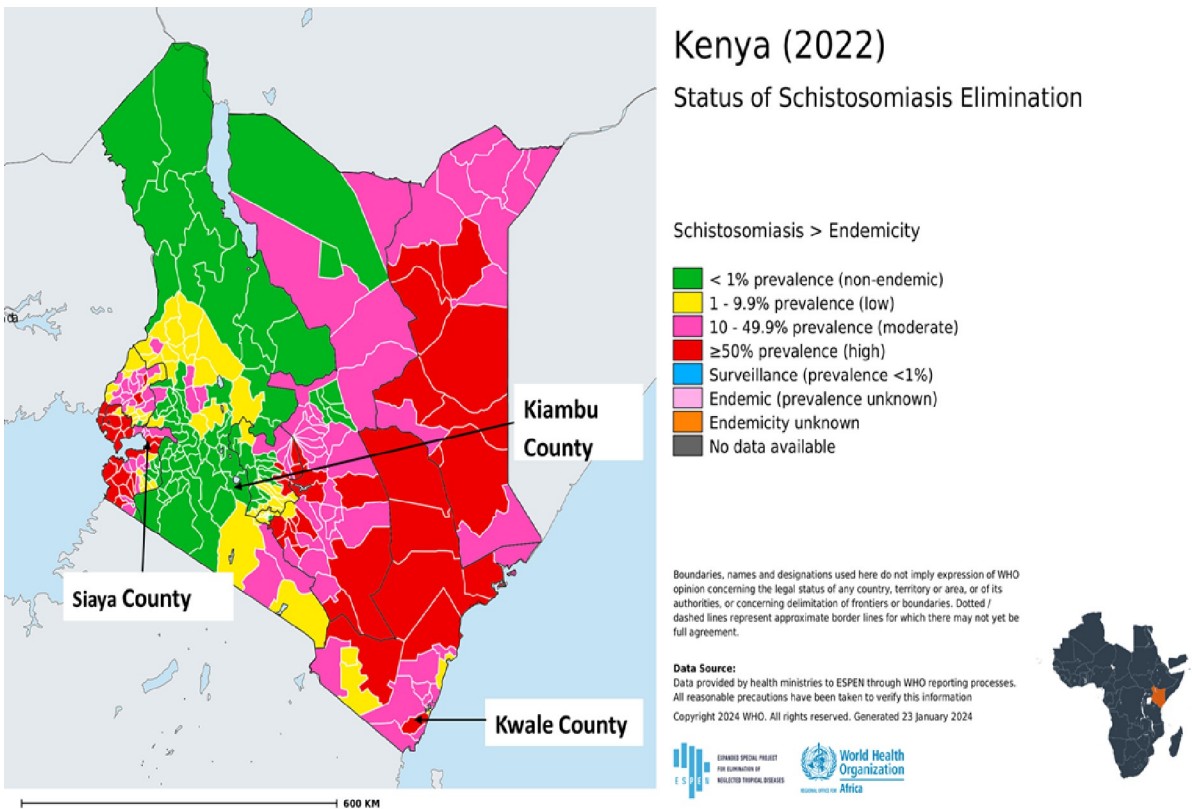

**Fig 1. The location of the three study sites.**

humid climatic conditions. The mean rainfall range is between 800–1600 mm annually. The second site was Mkongani Ward in Mwaluphamba location in Matuga Sub-county, Kwale County in the coastal region and endemic for *S. haematobium* [31–33]. Subsistence farming is the main economic activity of this area. The area is hot and humid all year round with annual mean temperatures range of 22˚C—34˚C, average relative humidity range of 70% - 80%, and annual rainfall range of 900–1500 mm. Altitude ranges from 0 to 462 meters above sea level. A third site that is non-endemic for schistosomiasis (acted as a control site) was involved in the current study [32]. This site was Kijabe Ward in Lari Sub-county, Kiambu County [34]. This site is largely mountainous and lies on the windward side of the Aberdare ranges in central Kenya. The main economic activity of this area is predominantly agriculture and subsistence farming. The area experiences relatively cold weather and is considerably rainy throughout the year with a mean annual rainfall of 1200mm.

## Study population

Study population comprised of 351 and 348 participants from the *S. mansoni* and the *S. haematobium* endemic sites, respectively. Additionally, 76 participants were recruited from the schistosomiasis non-endemic site and each provided both urine and stool samples thus acting as the control site for both the *S. mansoni* and *S. haematobium* endemic sites (Fig 2). However, one enrolled individual from the non-endemic site, a young boy, who had travelled to the Western Kenya region where SCH is endemic was removed from the analysis.

**Study samples.** This evaluation was done using microscopically confirmed samples collected from *S. mansoni* and *S. haematobium* endemic sites for three consecutive days in Kenya. In addition, both stool and urine samples from a *Schistosoma* non-endemic control site were similarly analyzed (Fig 2). Collection of the samples in the three study sites was done from February–June 2021. The stool samples were preserved in absolute ethanol at 1:3 mass to volume ratio, while 1 ml of the neat urine samples was aliquoted and stored for qPCR assay. These samples were stored at—20 ⁰C at a local laboratory in the field and later transported in dry ice to the Eastern and Southern Africa Centre of International Parasite Control (ESACIPAC)

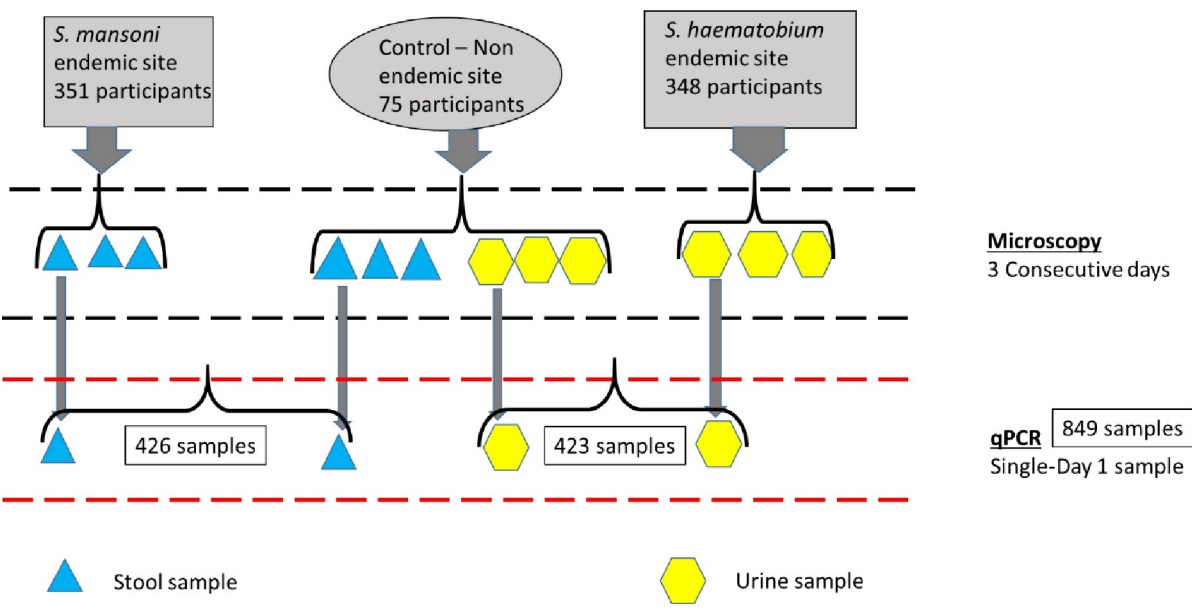

**Fig 2. Schematic diagram of the study population and procedures.**

laboratory in KEMRI, Nairobi where they were stored in a—40 ⁰C freezer until DNA extraction and qPCR assay was done.

**Microscopy.** Duplicate Kato-Katz thick smears (41.7mg) were made from the same stool sample in laboratories identified in the field sites and the smears examined under the microscope at 100x magnification [12]. *Schistosoma mansoni* eggs were counted and intensity expressed as eggs per gram of stool (epg) by multiplying the arithmetic mean of egg counts from the six slides per participant by 24 and categorized as per the WHO guidelines (S1 Appendix). The urine samples were also collected for three consecutive days, filtered and examined for ova of *S. haematobium*. The intensity of infection was calculated as the arithmetic mean of the eggs per 10 ml of urine from the total slides per participant and expressed as the numbers of eggs per 10 ml of urine as per the WHO guidelines (S1 Appendix) [35–37]. For quality assurance purposes, a random examination of 10% of the slides were re-examined by a senior laboratory technologist.

## DNA extraction

The DNA was isolated from all samples using QIAamp DNA mini kit (QIAGEN, Hilden, Germany) and as described by Aryeetey *et al* and Verweij *et al* with slight modifications [25, 38]. For stool, two hundred and fifty microlitres (250μl) of stool suspended in ethanol was added into 200μl of 2% polyvinylpolypyrrolidone (PVP) dissolved in 1X Phosphate Buffered Saline (PBS) solution. The mixture was homogenized using a vortex mixer for 1 minute and frozen overnight at -20˚C. The sample was then incubated for 10 minutes at 100˚C followed with a one-minute spin at 8000 rpm. After this, 200μl of tissue lysis buffer (ATL) containing 20μl of proteinase K was added and incubated for 2 hours at 55˚C. As an internal DNA extraction control, 1 μl of pDMD801 plasmid at a concentration of 100 pg/μl was added to the samples as a means of verifying the success and consistency of DNA extraction [39, 40]. A negative extraction (nuclease free water only) was included in every extraction cycle. For the urine samples, a similar procedure of DNA extraction was performed but using 200μl of the specimen, no resuspension in PVP solution and without an overnight freezing incubation step. The DNA from both urine and stool were finally double eluted in a volume of 50μl elution (AE) buffer. The DNA extraction and amplification processes were performed in separate rooms to prevent contaminations.

## Polymerase chain reaction

Parallel real-time qPCR assays were run for the *Schistosoma* genus and for the internal control. A *Schistosoma* qPCR was done using genus-specific primers amplifying a 77-basepair (bp) fragment of the internal transcribed spacer-2 (ITS2) subunit as described by Obeng and others with slight modification [41]. *Schistosoma* genus-specific primers Ssp48F (5'-GGT CTA GAT GAC TTG ATT GAG ATG CT-3') and Ssp124R (5'-TCC CGA GCG TGT ATA ATG TCA TTA-3') were used to amplify the 77-bp fragment of ITS2 and the amplification product detected by the probe, Ssp78T [FAM-5'-TGG GTT GTG CTC GAG TCG TGG C-3'-Black Hole Quencher] (Biolegio, Nijmegen, The Netherlands). The amplification was performed in a 7μl reaction mixture containing qPCR buffer (TaqPath ProAmp mastermix; Thermo Fisher), 12.5pmol of each *Schistosoma* genus-specific primer, 2.5pmol of the *Schistosoma* genus-specific double-labeled probe, and 2μL of the DNA sample using a StepOnePlus™ real-time qPCR System (Thermo Fisher Scientific Inc., USA). The thermocycler cycling conditions were as follows: 15 minutes at 95˚C, followed by 50 cycles, each of 15 seconds at 95˚C, 30 seconds at 60˚C, and 30 seconds at 72˚C. The qPCR output was reported as cycle threshold (Ct) value which represents amplification cycle in which fluorescent level exceeded the background

fluorescence. An internal control was also run per sample prior to the *Schistosoma* assay to test for successful DNA extraction and to ensure exclusion of any presence of qPCR inhibitors. Amplification of internal control (pDMD801 plasmid) was performed in a 7μl reaction mixture containing qPCR buffer (TaqPath ProAmp mastermix; Thermo Fisher), 12.5pmol of each pDMD801 plasmid primer (F: 5'-CTAACCTTCGTGATGAGCAATCG-3', R: 5'-GATCAGCTA CGTGAGGTCCTAC-3', 2.5pmol of the pDMD801 plasmid double-labeled probe:—56FAM/ AGCTAGTCG/ZEN/ATGCACTCCAGTCCTCCT/3IABkFQ/-3'), and 2μL of the DNA sample. The thermocycler used was set to give 15 minutes at 95˚C, followed by 40 cycles, each of 15 seconds at 95˚C, 30 seconds at 59˚C, and 30 seconds at 72˚C. Any sample that failed to amplify the internal control was re-extracted and the resulting DNA used as the template in the repeat qPCR. Preparation of master mix and addition of DNA template were done in separate laminar flow chambers which had previously been irradiated for 10 minutes using ultraviolet light and further sterilized using freshly made 10% bleach. The qPCR results were stratified into high (Ct < 30), moderate ($30 \leq Ct \geq 35$), low (Ct > 35) and negative (No amplification Ct = 0) [25].

## Data analysis

The Bayesian Latent Class Model (BLCM) was used for the data analysis. The first step was to generate a dichotomous variable "location" based on where the study participants were drawn from, that is, either the endemic or the non-endemic regions. The BLCM was then fit in Open Bugs version 3.2.2 software [42] and called from R software via B Rugs Package [43] to generate prevalence, sensitivity (Se), specificity (Sp) and predictive values of the two diagnostic assays for *Schistosoma* infection and species specific infections of *S. haematobium* and *S. mansoni*.

There are three key assumptions when constructing BLCM [44]. Firstly, the target population should consist of two or more sub-populations with different disease prevalence estimates. In this regard, the target population consisted of two separate subpopulations: endemic and non-endemic. The two sub-populations are either endemic for *S. haematobium* or *S. mansoni* or non-endemic [45]. Secondly, the sensitivity and specificity should be constant across subgroups. Finally, the tests should be conditionally independent given the disease status. On this point, conditional independence was assumed.

In our analysis, each sub-population was treated as a separate population k with its own prevalence ($P_k$). Each sub-population was subjected to two diagnostic tests, i (i = 1,2) The test results are distributed according to a multinomial model. The multinomial probabilities are defined using the specific test characteristics, that is, sensitivity and specificity and prevalence for each sub-population.

From the two sub-populations, the BLCM contained six parameters, that is, the sensitivity and the specificity of the two tests and the prevalence in each sub-population. These six parameters were estimated from the six degrees of freedom obtained from each of the two sub-populations. As per available literature, the Se and Sp estimates of microscopy for *S. mansoni* fall within the ranges (74.1%-89.7%) and (72.8%-100%), respectively [11, 46, 47]. The Se and Sp estimates of microscopy for *S. haematobium* fall within the ranges (69.9%-98%) and (85.6%-100%), respectively [10, 48]. These test ranges were used to specify the uniform prior distribution for the microscopy test within the BLCM. Non-informative priors (beta (1,1)) were utilized for the remaining parameters since no prior information was available for qPCR because of the innovativeness of the assay. The diagnostic performance of the tests was further assessed in terms of positive predictive value (PPV) and negative predictive value (NPV). The test estimates of sensitivity and specificity and the posterior distribution of the prevalence of the sub-

populations were reported as the median value with the associated posterior credible interval (PCI). A detailed description of the model is provided in the S2 Appendix.

## Results

### Demographics of study participants

Overall, a total of 849 samples were collected from schistosomiasis endemic and non-endemic (control site) sites. With regard to *S. haematobium* endemic site, a total of 348 samples were collected (Table 1). Regarding *S. mansoni* endemic site, 351 samples were collected. In terms of age in *S. mansoni* endemic population, nearly half of the participants 175 (49.9%) were aged between 5 to 15 years. The same trend was observed for *S. haematobium* endemic population at 179 (51.4%). A total of 150 samples were collected from the control site, majority of the participants 72 (48.0%) were between the age of 5 to 15 years. Gender distribution was balanced at 76 (50.7%) males and 74 (49.3%) females.

### Stool and urine examinations

The proportion of positive results by microscopy in the two endemic regions combined was 289/699 (41.3%). In terms of species-specific *Schistosoma* infection, the positivity for *S. haematobium* infection was 38.5% (134/348) and 44.2% (155/351) for *S. mansoni* infection. The overall intensity of *Schistosoma* infection by microscopy was predominantly light for both *S. mansoni* and *S. haematobium*. Out of the 155 cases with *S. mansoni* in the endemic population 26.8% (n = 94), 13.1% (n = 46) and 4.3% (n = 15) had light, moderate and heavy infections, respectively. With regard to *S. haematobium* in the endemic population, 26.1% (n = 91) had light infection and 12.4% (n = 43) had heavy infection. In the non-endemic population, all samples tested negative using microscopy for either *S. mansoni* or *S. haematobium* infection.

Using qPCR, in the two endemic sites altogether, 50.4% (352/699) were found to be positive. Regarding endemic sites, the positivity was 57.8% (203/351) and 42.8% (149/348) in the *S. mansoni* and *S. haematobium* sites, respectively. With regard to qPCR results for the endemic population, high intensity infections were 41.6% (n = 146) and 31.3% (n = 109), and low intensity infections were 6.3% (n = 22) and 5.2% (n = 18) for *S. mansoni* and *S. haematobium* sites respectively (Table 2).

**Table 1. Demographics of participants by study population.**

| Participant characteristics | Overall (N = 849) (%) | Endemic population | | Control site |
|---|---|---|---|---|
| | | *S mansoni* (n = 351) (%) | *S haematobium* (n = 348) (%) | (n = 150) (%) |
| **Age categories** | | | | |
| 5–15 years | 426 (50.2%) | 175 (49.9%) | 179 (51.4%) | 72 (48.0%) |
| 16–36 years | 135 (15.9%) | 60 (17.1%) | 63 (18.1%) | 12 (8.0%) |
| 27–37 years | 119 (14.0%) | 61 (17.4%) | 42 (12.1%) | 16 (10.7%) |
| 38–48 years | 78 (9.2%) | 26 (7.4%) | 30 (8.6%) | 22 (14.7%) |
| 49–59 years | 38 (4.5%) | 13 (3.7%) | 15 (4.3%) | 10 (6.6%) |
| 60 and above years | 53 (6.2%) | 16 (4.6%) | 19 (5.5%) | 18 (12.0%) |
| **Gender** | | | | |
| Male | 498 (58.7%) | 145 (41.3%) | 130 (37.4%) | 76 (50.7%) |
| Female | 351 (41.3%) | 206 (58.7%) | 218 (62.6%) | 74 (49.3%) |

**Table 2. Positivity and intensity of infection categories of *Schistosoma* infection.**

| Diagnostic test and intensity category | Overall | Endemic | Non-endemic |
|---|---|---|---|
| **Microscopy–*S. mansoni*** | | | |
| Negative | 271 (63.6%) | 196 (55.8%) | 75 (100%) |
| Light (1–99 EPG) | 94 (22.1%) | 94 (26.8%) | 0 |
| Moderate (100–399 EPG) | 46 (10.8%) | 46 (13.1%) | 0 |
| Heavy (≥400 EPG) | 15 (3.5%) | 15 (4.3%) | 0 |
| **Microscopy–*S. haematobium*** | | | |
| Negative | 289 (68.3%) | 214 (61.5%) | 75 (100%) |
| Light (1–50 eggs/10ml of urine) | 91 (21.5%) | 91 (26.1%) | 0 |
| Heavy (≥50 eggs/10ml of urine) | 43 (10.2%) | 43 (12.4%) | 0 |
| **qPCR–*S. mansoni*** | | | |
| Negative (No amplification) | 223 (52.3%) | 148 (42.2%) | 75 (100%) |
| High (Ct < 30) | 146 (34.3%) | 146 (41.6%) | 0 |
| Moderate (30 ≤Ct ≥ 35) | 35 (8.2%) | 35 (9.9%) | 0 |
| Low (Ct > 35) | 22 (5.2%) | 22 (6.3%) | 0 |
| **qPCR–*S. haematobium*** | | | |
| Negative (No amplification) | 274 (64.8%) | 199 (57.2%) | 75 (100%) |
| High (Ct < 30) | 109 (25.8%) | 109 (31.3%) | 0 |
| Moderate (30 ≤Ct ≥ 35) | 22 (5.2%) | 22 (6.3%) | 0 |
| Low (Ct > 35) | 18 (4.2%) | 18 (5.2%) | 0 |

## Test outcomes of *Schistosoma* infection

Tables 3 and 4 display the cross-tabulated performance of qPCR and microscopy for the diagnosis of *Schistosoma* infection by population, sites and species. Out of 348 samples from the *S. haematobium* endemic site, 128 tested positive by both the qPCR and microscopy assays. Regarding *S. mansoni* infection, a total of 137 individuals out of 351 tested positive by both the qPCR and microscopy assays in the endemic population. For the combined sites, 265 samples were positive while 304 were negative by both qPCR and microscopy respectively.

## Sensitivities and specificities of the diagnostic assays

In regards to *S haematobium* infection, the Se of microscopy (86.8% (PCI: 80.1, 93.1)) was higher than the Se of qPCR (84.5% (PCI: 77.8, 90.9)). As for the Sp, microscopy and qPCR were numerically similar at (99.1% (PCI: 95.2, 100.0)). On the other hand, for *S mansoni*, the

**Table 3. Cross-tabulated results for combination of qPCR and microscopy for the diagnosis of *Schistosoma* infection by population, site and species.**

| Population | Test outcome (qPCR; microscopy) | | | | Total (%) |
|---|---|---|---|---|---|
| | (++) | (+-) | (-+) | (—) | |
| ***S. haematobium*** | | | | | |
| Endemic site | 128 | 21 | 25 | 174 | 348 (41.0%) |
| ***S. mansoni*** | | | | | |
| Endemic site | 137 | 66 | 18 | 130 | 351 (41.3%) |
| Non-endemic (Control site) | 0 | 0 | 0 | 150 | 150 (17.7%) |
| Total | 265 | 87 | 43 | 454 | 849 (100%) |

+ Positive;—Negative

**Table 4. Overall cross-tabulated results for qPCR and microscopy for the diagnosis of *Schistosoma* infection.**

| Population | Test outcome (qPCR; Microscopy) | | | | Total (%) |
|---|---|---|---|---|---|
| | (++) | (+-) | (-+) | (—) | |
| Endemic site(s) (Combined) | 265 | 87 | 43 | 304 | 699 (82.3%) |
| Non-endemic (Control site) | 0 | 0 | 0 | 150 | 150 (17.7%) |
| Total | 265 | 87 | 43 | 454 | 849 (100%) |

+ Positive;—Negative

Se of qPCR (88.0% (PCI: 83.5, 89.6)) was higher than the Se of microscopy (75.0% (PCI: 74.1, 78.1)). As for the Sp microscopy and qPCR estimates were (99.3% (PCI: 96.5, 100.0)) and (98.3% (PCI: 92.2, 99.9)) respectively (Table 5).

## Overall sensitivities and specificities of the diagnostic assays

On evaluating the overall performance of the two techniques in the detection of genus *Schistosoma*, the Se of qPCR (86.4% (PCI: 82.1, 90.3)) was higher than the Se of microscopy (75.6% (PCI: 71.1, 80.0)). As for the Sp, microscopy and qPCR were numerically similar at (99.5% (PCI: 97.6, 100.0)). Strikingly, test parameters for specific sites were comparable to the overall performances except for sensitivity in which the overall sensitivity for microscopy technique was inferior to microscopy in the *S. haematobium* site.

## Predictive values and prevalences

The true positivity of *Schistosoma* infection determined using the BLCM by species varied in the two populations (Table 6). The estimated true positivity of *S haematobium* infection in the endemic population was recorded at 49.8% (PCI: 43.6, 55.6) which is higher than in the non-endemic population which was recorded at 0.9% (PCI: 0.0, 4.9). In the endemic population, qPCR had a PPV of 98.9% and an NPV of 86.5%, while microscopy had a PPV of 98.9% and an NPV of 88.3%. Conversely, in the non-endemic population, qPCR had a PPV of 46.4% and an NPV of 99.9%, while microscopy had a PPV of 46.5% and an NPV of 99.9% (Table 6).

In the endemic population, the recorded true positivity of *S. mansoni* was 63.5% (PCI: 57.4, 69.1), compared to 0.9% (PCI: 0.0, 4.9) observed in the non-endemic population. Within the endemic population, qPCR had a PPV of 98.9% and an NPV of 82.2%, while microscopy had a PPV of 99.5% and an NPV of 69.6%. Conversely, in the non-endemic population, qPCR demonstrated a low PPV of 32.8% and a high NPV of 99.9%, while microscopy showed a PPV of 50.5% and an NPV of 99.8% (Table 6).

**Table 5. Estimates of the sensitivity and specificity of PCR and microscopy and their respective 95% posterior credibility interval (PCI) and Youden indices for diagnosis of Schistosoma per site.**

| Test parameter [1] | *S. haematobium* | | *S. mansoni* | |
|---|---|---|---|---|
| | Estimate (95% PCI) [2] | Youden indices | Estimate (95% PCI) [2] | Youden indices |
| $Se_{qPCR}$ | 84.5 (77.8,90.9) | Youden index$_{qPCR}$ 83.6 | 88.0 (83.5,89.6) | Youden index$_{qPCR}$ 86.3 |
| $Sp_{qPCR}$ | 99.1 (95.2,100.0) | | 98.3 (92.2,99.9) | |
| $Se_M$ | 86.8 (80.1,93.1) | Youden index$_M$ 85.9 | 75.0 (74.1,78.1) | Youden index$_M$ 74.3 |
| $Sp_M$ | 99.1 (95.2,100.0) | | 99.3 (96.5,100.0) | |

1 Median estimate; 2 Posterior Credible Interval; M-Microscopy; Se- Sensitivity; Sp-Specificity

**Table 6. Estimates of the predictive values of qPCR and microscopy and their respective 95% posterior credibility interval (PCI) for diagnosis of *Schistosoma* infection.**

| Population | Test Parameter [1] | *S. haematobium* Estimate (95% PCI) [2] | | *S. mansoni* Estimate (95% PCI) [2] | |
|---|---|---|---|---|---|
| | | qPCR | Microscopy | qPCR | Microscopy |
| Endemic | PPV | 98.9% (94.0, 100.0.) | 98.9%(94.2, 100.0) | 98.9%(94.5, 100.0) | 99.5%(97.3, 100.0) |
| | NPV | 86.5%(80.2, 92.5) | 88.3%(82.0, 94.1) | 82.2%(75.4, 86.4) | 69.6%(63.9, 75.5) |
| | Positivity | 49.8%(43.6, 55.6) | | 63.5%(57.4, 69.1) | |
| Non-endemic | PPV | 46.4%(2.2, 97.0) | 46.5%(2.2, 97.1) | 32.8% (1.3, 94.7) | 50.5%(2.6, 97.6) |
| | NPV | 99.9%(99.2, 100.0) | 99.9%(99.3, 100.0) | 99.9%(99.3, 100.0) | 99.8%(98.7, 100.0) |
| | Positivity | 0.9%(0.0, 4.9) | | 0.9%(0.0, 4.9) | |

1 Median estimate; 2 Posterior Credible Interval; PPV- Positive Predictive Value; NPV- Negative Predictive Value

## Discussion

This study presents the performance of microscopy test on urine and stool samples collected over three consecutive days versus genus-specific *Schistosoma* qPCR assay done on a day one sample only. The analysis of performance of the tests in this study was done using BLCM. This model is primarily based on the fact that there is no reference test and involves explicit definition of the conditions targeted by the tests being evaluated and the statistical model's complexity [49]. Traditionally methods of estimating sensitivity and specificity are based on the premise that there is a gold standard or reference tests [50]. A drawback of this approach is that, given the imperfection of these reference tests the results generated may be prone to misclassification errors. On the other hand, BLCM allows estimation of the sensitivity and specificity without assumptions of the true disease status of each individual [49, 51]. Additionally, the model allows incorporation of prior information from literature about parameters such as sensitivity, specificity or prevalence enhancing the estimates generated. The defined conditions in the current study is that stool samples collected from *S. mansoni* endemic and urine samples from *S. haematobium* endemic areas were combined into one single group defined as endemic group while the paired (stool and urine) samples collected from the control sites defined as non-endemic group. Additionally, species-specific results were also reported.

This study shows a higher Genus *Schistosoma* detection rate by qPCR of 50.4% compared to microscopy rate that was 41.3% in endemic areas. These findings are despite the repeated microscopy in this study that was meant to cure microscopy's documented low sensitivity which is possibly attributed to egg output variations and uneven distribution of eggs in the sample [52]. The superior performance of the qPCR is in agreement with other similar studies in different regions [20–23, 28]. This observation has previously led to qPCR assay being fronted as a powerful tool for accurate establishment of prevalence and intensity of *Schistosoma* infection especially in monitoring the effectiveness of programmatic directed treatment [25]. Besides the superior performance of the qPCR assay, the current study presents an additional advantageous platform in that it enables detection of the schistosomiasis infection without subtyping the amplicon to the species level, which is not necessary in the context of control programmes. Moreover, this assay has also simplified what would otherwise be a laborious process of analyzing a single type of sample, either stool or urine, separately using different protocols.

The current study shows specificity of 99.5% (PCI: 97.6–100.0) for qPCR and microscopy and sensitivity of 86.4% (PCI: 82.1–90.3) for qPCR, 75.6% (PCI: 71.1–80.0) for microscopy.

According to the WHO TPP guidelines for schistosomiasis diagnostics, the ideal requirements for the clinical sensitivity and clinical specificity should be above 75% and 96.5% respectively for a sample size of 100 individuals [4]. In this study, the specificity target was exceedingly attained by both tests. However, overall sensitivity of qPCR was much higher while performance by the microscopy was close to cut-off of the WHO-recommended values. Interestingly microscopy showed higher sensitivity in *S. haematobium* compared *to S. mansoni* and in relations to the above WHO TPP. It is worth noting that urine filtration and urine PCR utilized 10mls and 1ml of neat urine samples respectively. Conversely, Kato-Katz utilized 41.7g of stool while stool PCR utilized sample preserved in absolute ethanol at 1:3 mass to volume ratio. This could probably have effects on the recovery of eggs and detection of the infection in favour of *S. haematobium* microscopy due to samples volumes compared to its PCR and both microscopy and PCR in *S. mansoni*.

The low sensitivity due to low detection limit associated with microscopy in *S. mansoni* has been observed elsewhere and shown to hamper its use in the settings with light intensity infections. This could potentially have an impact on control programmes by inhibiting confirmation of cure rate after successful treatment and underestimating true prevalence especially in low infection settings [37, 53, 54]. This provides additional justification for use of other sensitive diagnostic assays alongside microscopy in the *S. mansoni* settings. This study also shows an improvement in the performance of microscopy in comparison to another study by Aryeetey *et al.* [25]. Notably, the study by Aryeetey observed that the poor sensitivity by microscopy could have been caused by PCR sensitivity overestimate and thus recommended using BLCM, which does not use an apparent "true gold standard" for further work as was done in this study. Elsewhere, using this model, a study done on school children in Tanzania found out that Kato-Katz similarly had lower sensitivity of 89.7% compared to real-time PCR whose sensitivity was 98.7% [11]. However, the improved performance in this current study could probably be attributed to examination of samples over three consecutive days.

The results show discordance between the qPCR and microscopy. Eighty-seven samples were positive by qPCR but negative by microscopy. Similarly, there was increase in the numbers of samples with heavy infections category and overall reduction in number of negative individuals after analysis by qPCR. This could be an affirmation of the superior detection capacity of the qPCR assay even in the event of low egg count in the individuals [28, 55]. Also, this could be due to the fact that probably the qPCR systems can detect infection during larval stage or before eggs are found in stool [21]. Further, ability of the qPCR to at times to pick *S. haematobium* DNA present in stool samples may yield more positives [18]. On the flipside, 43 samples were negative by qPCR and positive by microscopy. A similar observation was observed in a study by Allam *et al* [55]. This could be as a result of diverse reasons. First, microscopy was done on samples collected on three consecutive days while qPCR was done on day one samples, a limitation acknowledged by this study due to associated cost and storage challenges that would accompany increased number of aliquots. This could have increased sensitivity in microscopy and not PCR in the event of variation in egg shedding across the three days. Secondly, detection of DNA could fail due to the presence of PCR inhibition. Similar observations have been documented in other studies [7, 38]. However, PCR inhibition could be ruled out in the current study since a separate and parallel assay for internal control was ran for all the samples. This notwithstanding, qPCR has additional advantages over microscopy in that its performance is not subject to methodological variability like microscopy quality and could be done in centralized facility for accurate monitoring diseases control activities [56]

## Study limitations

The overarching limitation of this study is that qPCR assay was done on the samples collected on day one only which could potentially affect positivity in the context of varying egg shedding in the study participants. Further parallel analysis using microscopy and qPCR for samples collected at the identical time points over three consecutive days is recommended to enrich the body of knowledge on the performance of the two assays. Additionally, applying molecular techniques such as qPCR in community settings can be challenging due to potential limitations occasioned by high costs and significant resource requirements.

## Conclusions

A modified genus *Schistosoma* assay qPCR is a overally sensitive technique for the detection of schistosomiasis compared to microscopy and thus suitable for use in areas with low infection due to control interventions. Additionally, this assay provides a common integrated platform for monitoring of the progress of elimination efforts in both the *S. mansoni* and *S. haematobium* endemic areas. It is therefore recommended for use by the national programmes for accurate determination elimination endpoints and disease surveillance to avert the re-emergence of the infection after effective control. However, to mitigate the cost, a subset of the sample size could be analyzed using the assay as a proxy indicator of the infection level.

## Supporting information

**S1 Appendix. WHO intensity thresholds for light, moderate and heavy infections for Schistosoma.**
(DOCX)

**S2 Appendix. BLCM used for data analysis.**
(DOCX)

## Author Contributions

**Conceptualization:** Henry Kanyi, Thierry Ramos, Sarah Nogaro, Sammy M. Njenga.

**Data curation:** Henry Kanyi.

**Formal analysis:** Henry Kanyi, Richelle W. Kihoro, Benard Chieng, Sylvie Araka, Henry Emisiko.

**Investigation:** Sammy M. Njenga.

**Methodology:** Henry Kanyi, Sarah Nogaro, Sammy M. Njenga.

**Project administration:** Henry Kanyi, Sammy M. Njenga.

**Supervision:** Henry Kanyi, Sarah Nogaro, Sammy M. Njenga.

**Validation:** Henry Kanyi.

**Writing – original draft:** Henry Kanyi, Richelle W. Kihoro, Sarah Nogaro, Sammy M. Njenga.

**Writing – review & editing:** Henry Kanyi, Richelle W. Kihoro, Sarah Nogaro, Sammy M. Njenga.

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
