## [Decision Letter · Decision Letter 0]

31 May 2024

PONE-D-24-12369Evaluation of a modified quantitative polymerase chain reaction assay for genus Schistosoma detection using stool and urine samples from schistosomiasis endemic areas in KenyaPLOS ONE

Dear Dr. Kanyi,

Thank you for submitting your manuscript to PLOS ONE. After careful consideration, we feel that it has merit but does not fully meet PLOS ONE’s publication criteria as it currently stands. Therefore, we invite you to submit a revised version of the manuscript that addresses the points raised during the review process.

We look forward to receiving your revised manuscript.

Kind regards,

Raquel Inocencio da Luz, Phd

Academic Editor

PLOS ONE

Journal Requirements:

2. Funding Information and Financial Disclosure sections do not match:

""We note that the grant information you provided in the ‘Funding Information’ and ‘Financial Disclosure’ sections do not match. 

3. In the online submission form, you indicated that "All relevant data are within the manuscript and its Supporting Information files. The raw data will be available upon request"

4. Please ensure that you refer to Figure 1 in your text as, if accepted, production will need this reference to link the reader to the figure.

Reviewers' comments:

Reviewer's Responses to Questions

**Comments to the Author**

1. Is the manuscript technically sound, and do the data support the conclusions?

Reviewer #1: Partly

Reviewer #2: Yes

2. Has the statistical analysis been performed appropriately and rigorously? 

Reviewer #1: Yes

Reviewer #2: Yes

3. Have the authors made all data underlying the findings in their manuscript fully available?

Reviewer #1: No

Reviewer #2: Yes

4. Is the manuscript presented in an intelligible fashion and written in standard English?

Reviewer #1: Yes

Reviewer #2: Yes

5. Review Comments to the Author

Reviewer #1: This is an interesting study pointing towards the need of sensitive diagnostic tools such as qPCR assays particularly in disease control and elimination process. Refer to the specific comments given within the attached manuscript embedded in Comments boxes, to improve the presented work further.

Reviewer #2: The manuscript by Kanyi et al., "Evaluation of a modified quantitative polymerase chain reaction assay for genus Schistosoma detection using stool and urine samples from schistosomiasis endemic areas in Kenya", describes the development of a modified singleplex Schistosoma genus-specific quantitative polymerase chain reaction (qPCR) assay for Schistosoma diagnosis. The manuscript is well written, and the results align with the conclusions. However, I have the following comments that need to be looked at:

1. It would be important for the authors to mention the chemotherapeutic strategies that have been deployed for schistosomiasis control and any successes that have been achieved using this strategy in the introduction section of the manuscript

2. In lines 86–92, where the authors highlight the limitations of current diagnostic tests, the requirement for well-trained and experienced personnel is mentioned. I am wondering if the qPCR test developed in this manuscript would solve this problem. In addition, how suitable is the developed test for low-resource settings? In the same light, the authors mention qPCR as being advantageous over conventional PCR (lines 117–119), but what about the sophistication and equipment?

3. Concerning the ethical considerations, the authors mention that participants provided written informed consent. Here are my concerns about this: Were all participants above the age of consent? Were all participants literate?

How was consent obtained from the illiterate and minors? More clarity is needed on this.

4. Concerning the study sites selected, the authors should provide any previous data on Schistosoma prevalence. In addition, the authors should provide consistent data on the different study sites (e.g., rainfall, economic activity, humidity, temperature, etc)

5. Figure 1 should read, "Map of Kenya...", and not "Kenyan map...". In addition, I am wondering if permission is needed from ESPEN for this map to be used. In addition, I am wondering if there is no prevalence data that is more recent, given that this map was produced 6 years ago.

6. It would be important for the authors to provide information on other diagnostic parameters, like likelihood ratios, % positive agreement, Youden's index, etc).

7. In lines 435–437, the authors should provide specific sensitivity information on the study being referred to.

8. Lastly, the manuscript needs to be carefully proofread again and grammatical errors corrected. DNA (line 112) is an acid, the full meaning of NTD (line 146) should be provided. A consistent style should be used to indicate temperatures

6. PLOS authors have the option to publish the peer review history of their article (what does this mean?). If published, this will include your full peer review and any attached files.

Reviewer #1: No

Reviewer #2: **Yes: **Robert Adamu Shey

---

## [Author Response · Author response to Decision Letter 0]

30 Jul 2024

Racquel Inocencio da Luz, PhD

Academic Editor 

PLOS ONE

30th July 2024

Dear Dr. Racquel Inocencio da Luz,

Re: Revised version of manuscript reference PONE-D-24-12369 

Thank you for giving us an opportunity to revise our manuscript entitled " Evaluation of a modified quantitative polymerase chain reaction assay for genus Schistosoma detection using stool and urine samples from schistosomiasis endemic areas in Kenya " for submission to PLOS ONE. We are grateful for the valuable feedback from both you and the reviewers, which has helped us to enhance the paper. We are pleased to resubmit the article for your further consideration, incorporating changes based on your insightful suggestions. We believe our revisions and the responses provided below adequately address the issues and concerns raised

For your review here is a point by point response to the comments and questions raised. The comments are indicated in bold while our responses are not bold.

Journal Requirements

The manuscript has been reformatted as per the requirement of PLOS ONE’s requirements

2. Funding Information and Financial Disclosure sections do not match: ""We note that the grant information you provided in the ‘Funding Information’ and ‘Financial Disclosure’ sections do not match. When you resubmit, please ensure that you provide the correct grant numbers for the awards you received for your study in the ‘Funding Information’ section.

The text in the “Funding Information” and “Financial Disclosure” sections have been rewritten to match as follows: 

“This work was supported by the Bill and Melinda Gates Foundation (grant award: INV_024908) that was awarded to SH. The funder had no role in decision to publish, or preparation of the manuscript.”

3. In the online submission form, you indicated that "All relevant data are within the manuscript and its Supporting Information files. The raw data will be available upon request"

Kindly note that the consent forms utilized to collect data from participants explicitly stated that access to the data is limited to study staff, representatives from FIND and KEMRI, members of the ethics committee and the regulatory authorities. Therefore, access to anonymized and de-identified data can be obtained by emailing the Kenya Medical Research Institute (KEMRI) Scientific Ethics Review Unit (SERU) at seru@kemri.go.ke

4. Please ensure that you refer to Figure 1 in your text as, if accepted, production will need this reference to link the reader to the figure.

Reference to Figure 1 has been provided in lines 159 – 161. The figure has been updated. The source has also been acknowledged and the access date provided accordingly. 

5. Please include captions for your Supporting Information files at the end of your manuscript, and update any in-text citations to match accordingly.

Thank you for this comment we have provided captions at the end of the manuscript for the supporting information and have updated in- text citations to match accordingly (Line 677 - 680) & (Line 184 - 193 ;Line 248 - line 281)

Based on the recommendations from the reviewers some other references have been introduced and captured accordingly. 

Reviewer 1 Comments

1. It would be important for the authors to mention the chemotherapeutic strategies that have been deployed for schistosomiasis control and any successes that have been achieved using this strategy in the introduction section of the manuscript

Agreed, this has been added in the introduction section. A paragraph in the introduction section has been added to give context to the main chemotherapeutic strategy used for schistosomiasis through mass drug administration (MDA) of praziquantel. This strategy has mostly been implemented by the National School Based Deworming Programme (NSBDP) and has shown some impact on lowered the overall prevalence of schistosomiasis after repeated rounds of MDAs. The paragraph is on Line 64 - 70. 

2. In lines 86–92, where the authors highlight the limitations of current diagnostic tests, the requirement for well-trained and experienced personnel is mentioned. I am wondering if the qPCR test developed in this manuscript would solve this problem. In addition, how suitable is the developed test for low-resource settings? In the same light, the authors mention qPCR as being advantageous over conventional PCR (lines 117–119), but what about the sophistication and equipment?

We thank Reviewer 1 for the above comments. Please note that our recommendations are as follows (Line 468 – 471):- It is therefore recommended for use by the national programmes for accurate determination elimination endpoints and disease surveillance to avert the re-emergence of the infection after effective control. However, to mitigate the cost, a subset of the sample size could be analyzed using the assay as a proxy indicator of the infection level.

We are cognizant that cost is a limiting factor. However, the recommended test is not for routine use but rather for determination of the diseases elimination endpoints. Further, we recommend that a subset can be selected based on the available funding. This will enable striking of the balance between cost and accurate determination of the disease elimination.

3. Concerning the ethical considerations, the authors mention that participants provided written informed consent. Here are my concerns about this: Were all participants above the age of consent? Were all participants literate? How was consent obtained from the illiterate and minors? More clarity is needed on this. 

The points raised have valid concerns. We have added the following information in the ethics statement to clarify these matters. For participants less than 18 years old, parental consent was obtained, with assent from children (13- <18 years) before enrollment. Illiterate participants provided a thumbprint on the consent form and the consent form was also signed by an impartial witness. This information is provided in Line 133 - 136. 

4. Concerning the study sites selected, the authors should provide any previous data on Schistosoma prevalence. In addition, the authors should provide consistent data on the different study sites (e.g., rainfall, economic activity, humidity, temperature, etc)

In response to the comments above we have done the following:-

i. The three study sites have been amended to provide consistent information on the general location, economic activities, general climatic conditions and rainfall. Line 140 - 157

ii. We have provided the data supporting prevalences in the endemic sites and the reference for classification of the control site as non-endemic sites by the Ministry of Health, Kenya (Line 154)

iii. We have also provided a Kenyan map with updated endemicity for Schistosomiasis. In addition to the endemicity of the sites, the maps provide the location of the sites. Line 159 - 161

5. Figure 1 should read, "Map of Kenya...", and not "Kenyan map...". In addition, I am wondering if permission is needed from ESPEN for this map to be used. In addition, I am wondering if there is no prevalence data that is more recent, given that this map was produced 6 years ago. 

The most recent map (2022) has been prepared, renamed and acknowledged as per review comments. Line 159 -161. With regard to permission to use the map (Figure 1), we reached out to ESPEN and advised that no permission is required, but only to acknowledge the source and date accessed. 

6. It would be important for the authors to provide information on other diagnostic parameters, like likelihood ratios, % positive agreement, Youden's index, etc).

Thank you for this comment. As added in the discussion section they are two main methods of estimating sensitivity and specificity 1) method that utilizes a gold standard or reference test known as frequentists statistics 2) methods that do not utilize a gold standard known as Bayesian methods this is in Line 382 to 388. Bayesian statistics has been utilized in the data analysis of the paper. Cohen Kappa Index is a test of agreement that utilizes frequentist statistics. This also applies to likelihood ratios, % positive agreement but we have added Youden's index in table 4 of the manuscript in line 342 - 344

7. In lines 435–437, the authors should provide specific sensitivity information on the study being referred to. 

The specific sensitivity of the study being referred to has been added that is the Tanzanian study found a sensitivity for Kato-Katz at 89.7% and sensitivity for qPCR at 98.7%. This is in line 428 - 430

8. Lastly, the manuscript needs to be carefully proofread again and grammatical errors corrected. DNA (line 112) is an acid, the full meaning of NTD (line 146) should be provided. A consistent style should be used to indicate temperatures 

We thank Reviewer 1 for this comment. We have undertaken proofreading of the manuscript and corrected various grammatical errors throughout the manuscript. 

Reviewer 2 Comments

1. Abstract - Refer to the main text comments and apply the modifications here in to the abstract as well. 

We have made the necessary changes to the entire abstract

2. Abstract Introduction- Move to methodology

This sentence has been moved to methodology (Line 9 -11 moved to 16 -17)

3. Abstract Methodology- Any particular reason why S. japonicum (being a species causing significant morbidity in SEA) was not looked in to? 

This is because S. japonicum is not geographically distributed in the African continent and is predominantly found in Asia. There was therefore no site to collect site from in Kenya for comparison

4. Abstract Methodology - What types of samples?

The type of samples has been added in brackets for clarity. Please see line 17 – 18. 

5. Abstract Results - Not clear how the denominator becomes 849, and not 699. This should focus exclusively on the endemic regions.

Agreed and revised in line 21 -22. Please note that the study had 3 sites (two endemic and one non-endemic site) where samples were collected as follows: - In S. mansoni endemic site, 351 stool samples were collected while in S. haematobium endemic site 348 urine samples were collected. This added up to 699 samples in the two endemic sites. However, in non-endemic site, each of the 75 participants provided both stool and urine samples. Therefore, 150 samples (75 participants x 2 samples each) were collected in the non-endemic site. Overally then, 699 and 150 samples (total of 849) were collected. 

6. Abstract Results - Same as above

Agreed, this has been edited as indicated above

7. Abstract Results - Both microscopy and PCR is supposed to provide quantitative outcomes. A briefing of the comparison needs to come in here.

Comparison has been provided on the performance of the two techniques (Line 25 to 27).

8. Introduction - The diagnosis of schistosomiasis has been facilitated through the application of different PCR techniques, including conventional PCR, qPCR, as well as digital PCR (ddPCR). Suggest briefing with appropriate references.

Thank you for this important suggestion; a topic sentence has been added at the beginning of the paragraph to introduce the concept of PCR techniques in control of schistosomiasis. This is found in line 97 to 99. 

9. Introduction- Specify the study objectives

Agreed, this has been done on the last paragraph of the introduction (Line 121 to 124)

10. Introduction- Comment on line 108 – 114 Shouldn’t be places in the Introduction, and instead can be incorporated in to the discussion. – 

Thank you for this insightful comment. However, although the statement can also fit in the discussion, I strongly feel that it also fits here for two reasons. The first reason is that it lays basis for the reader to understand the background/general scope of the work reported in this manuscript. Secondly, it exemplifies the how the current PCR method was different from other PCR done on the schistosomiasis and specifically why it is referred to as “modified” under Line 116 - 121

11. Introduction- Not a abbreviation introduced above in the text

Defined – line 118

12. Materials and Methods- Study area- This section addresses both study type as well as the sites.

Agreed, the title of this section has been amended to reflect the information found in this section. This is in Line 138

13. Materials and Methods- Suggest citing the publication(s) arose from the primary study – Line 139 - 140

Although the samples utilized in this study are from stored, there is no publication that has been done on these samples. Therefore it is expected that this will be the first publication and that will be cited by any subsequent publications if any. 

14. Materials and Methods- The figure appears to be from another publication/source. Need to cite the source with permissions. 

Line 159 - 161. With regard to permission to use the map (Figure 1), we reached out to ESPEN and advised that no permission is required, but only to acknowledge the source. Same response provided to similar comment by reviewer 1 in comment 5. Thanks. 

15. Materials and Methods- Not abbreviated previously

Agreed, the text containing this abbreviation has been rephrased in line 167 – 170. The abbreviation was done in line 118

16. Materials and Methods- Were these microscopic findings published previously? Please cite and link if done so.

As indicated earlier the microscopy results are not yet published and thus this is first publication and which compares both the microscopy and PCR results. Line 173

17. Materials and Methods- Either briefly describe or cite a reference to provide the methodological aspects.

A citation is in place that contains the Kato-Katz technique used for stool examination (Reference Number 12) in line 187

18. Materials and Methods- Outcome of two slides? How were the samples from three days considered (was it altogether six smears)? Specify

Thank you for this comment, this has been clarified as follows:-. The samples were considered altogether from the six smears. A sample was considered to be positive if any of the slide was positive while a negative was where it was negative by all the slides. In the case of S. mansoni, eggs were counted and intensity expressed as eggs per gram of stool (epg) by multiplying the arithmetic mean of egg counts from the six slides per participant by 24 and categorized as per the WHO guidelines. In the case of S. haematobium the intensity of infection was calculated as the arithmetic mean of the eggs per 10 ml of urine from the six slides per participant and expressed as the numbers of eggs per 10 ml of urine as per the WHO guidelines, this can be found on Line 187 -193.

19. Materials and Methods- Specify how the quality assurance was ensured in microscopic analysis

Thank you for raising an important point. For quality assurance purposes, a random examination of 10% of the slides were re-examined by senior technologists. This is in Line 193 to 194

20. Materials and Methods- Specify the ‘modification

---

## [Decision Letter · Decision Letter 1]

26 Aug 2024

Evaluation of a modified quantitative polymerase chain reaction assay for genus Schistosoma detection using stool and urine samples from schistosomiasis endemic areas in Kenya

PONE-D-24-12369R1

Dear Dr. Henry Muriuki Kanyi,

We’re pleased to inform you that your manuscript has been judged scientifically suitable for publication and will be formally accepted for publication once it meets all outstanding technical requirements.

Kind regards,

Shawky M Aboelhadid, PhD

Academic Editor

PLOS ONE

Additional Editor Comments (optional):

Reviewers' comments:

Reviewer's Responses to Questions

**Comments to the Author**

1. If the authors have adequately addressed your comments raised in a previous round of review and you feel that this manuscript is now acceptable for publication, you may indicate that here to bypass the “Comments to the Author” section, enter your conflict of interest statement in the “Confidential to Editor” section, and submit your "Accept" recommendation.

Reviewer #1: All comments have been addressed

Reviewer #2: All comments have been addressed

2. Is the manuscript technically sound, and do the data support the conclusions?

Reviewer #1: Yes

Reviewer #2: Yes

3. Has the statistical analysis been performed appropriately and rigorously? 

Reviewer #1: Yes

Reviewer #2: Yes

4. Have the authors made all data underlying the findings in their manuscript fully available?

Reviewer #1: Yes

Reviewer #2: Yes

5. Is the manuscript presented in an intelligible fashion and written in standard English?

Reviewer #1: Yes

Reviewer #2: Yes

6. Review Comments to the Author

Reviewer #1: (No Response)

Reviewer #2: Thank you for addressing the comments and providing the reviewed version. I am satisfied with the issues that have been addressed.

7. PLOS authors have the option to publish the peer review history of their article (what does this mean?). If published, this will include your full peer review and any attached files.

Reviewer #1: No

Reviewer #2: **Yes: **Robert Adamu Shey

---

## [Editor Report · Acceptance letter]

11 Sep 2024

PONE-D-24-12369R1 

PLOS ONE

Dear Dr. Kanyi, 

I'm pleased to inform you that your manuscript has been deemed suitable for publication in PLOS ONE. Congratulations! Your manuscript is now being handed over to our production team.

Kind regards, 

on behalf of

Professor Shawky M Aboelhadid 

Academic Editor

PLOS ONE